# DJ-1 Alleviates Neuroinflammation and the Related Blood-Spinal Cord Barrier Destruction by Suppressing NLRP3 Inflammasome Activation via SOCS1/Rac1/ROS Pathway in a Rat Model of Traumatic Spinal Cord Injury

**DOI:** 10.3390/jcm11133716

**Published:** 2022-06-27

**Authors:** Lingxin Cai, Liansheng Gao, Guoqiang Zhang, Hanhai Zeng, Xinyan Wu, Xiaoxiao Tan, Cong Qian, Gao Chen

**Affiliations:** 1Department of Neurological Surgery, Second Affiliated Hospital of Zhejiang University School of Medicine, Hangzhou 310009, China; 22018421@zju.edu.cn (L.C.); 20918642@zju.edu.cn (L.G.); 22118147@zju.edu.cn (G.Z.); 11918362@zju.edu.cn (H.Z.); 11918535@zju.edu.cn (X.W.); tanxiaoxiao1988@163.com (X.T.); 2Key Laboratory of Precise Treatment and Clinical Translational Research of Neurological Diseases, Hangzhou 310009, China

**Keywords:** DJ-1, neuroinflammation, NLRP3 inflammasome, traumatic spinal cord injury

## Abstract

DJ-1 has been shown to play essential roles in neuronal protection and anti-inflammation in nervous system diseases. This study aimed to explore how DJ-1 regulates neuroinflammation after traumatic spinal cord injury (t-SCI). The rat model of spinal cord injury was established by the clamping method. The Basso, Beattie, Bresnahan (BBB) score and the inclined plane test (IPT) were used to evaluate neurological function. Western blot was then applied to test the levels of DJ-1, NLRP3, SOCS1, and related proinflammatory factors (cleaved caspase 1, IL-1β and IL-18); ROS level was also examined. The distribution of DJ-1 was assessed by immunofluorescence staining (IF). BSCB integrity was assessed by the level of MMP-9 and tight junction proteins (Claudin-5, Occludin and ZO-1). We found that DJ-1 became significantly elevated after t-SCI and was mainly located in neurons. Knockdown of DJ-1 with specific siRNA aggravated NLRP3 inflammasome-related neuroinflammation and strengthened the disruption of BSCB integrity. However, the upregulation of DJ-1 by Sodium benzoate (SB) reversed these effects and improved neurological function. Furthermore, SOCS1-siRNA attenuated the neuroprotective effects of DJ-1 and increased the ROS, Rac1 and NLRP3. In conclusion, DJ-1 may alleviate neuroinflammation and the related BSCB destruction after t-SCI by suppressing NLRP3 inflammasome activation by SOCS1/Rac1/ROS pathways. DJ-1 shows potential as a feasible target for mediating neuroinflammation after t-SCI.

## 1. Introduction

Traumatic spinal cord injury (t-SCI) caused by events such as a sports accident, traffic accident or a fall from height often results in persistent and severe motor and sensory dysfunction, leading to a reduced quality of life and a heavy medical burden on families and society [1]. Existing clinical treatments for t-SCI can partially reduce damage, but their long-term effects are limited [2]. Neuroinflammation is one of the essential pathological reactions after t-SCI and is regarded as a significant determinant of consequent neurological outcomes [3,4,5]. Previous studies have found that suppressing the neuroinflammation response could inhibit the generation and expansion of t-SCI induced secondary tissue damage to improve locomotor function recovery after t-SCI [6]. Therefore, it is necessary to further understand the complex inflammation cascade mechanism after t-SCI and find new targets for regulating neuroinflammation [7].

The DJ-1 gene is more expressed in patients with Parkinson’s disease (PD) and was originally regarded as a causative gene of PD [8]. DJ-1 participates in various pathophysiology activities, such as tumorigenesis, mitochondrial function regulation, and protein glycosylation inhibition [9,10]. Existing research shows that DJ-1 exhibits a nerve protective effect in neurodegenerative disease and cerebral ischemia [11,12,13], and that the potential mechanism may be associated with inflammation regulation [14]. DJ-1 exerts anti-inflammatory effects in ischemic stroke by inhibiting inflammatory cytokines TNF-α, interleukin (IL)-1β, and IL-18 [15,16]. Ya-Jie J et al. revealed that the down-regulation of DJ-1 accelerated microglia-mediated neuroinflammation in the pathogenesis of PD [17]. However, whether DJ-1 exerts protective effects on t-SCI by regulating inflammation remains unclear. Recently, nod-like receptor protein 3 (NLRP3) inflammasome has been suggested to play an essential role in neuroinflammatory mechanisms [18], promoting the maturation and release of IL-1β and IL-18 [19]. NLRP3 inflammasome was found to be activated after t-SCI, and inhibiting NLRP3 inflammasome activation can reduce neuroinflammatory response and promote nerve recovery [20,21].

We hypothesized that DJ-1 might mediate SCI-induced neuroinflammation to accelerate neurological function and improve clinical prognosis. Therefore, we established the rat t-SCI model in this study and used DJ-1 siRNA and DJ-1 agonist SB, to down-regulate and up-regulate DJ-1, respectively, suggesting that DJ-1 could reduce neuroinflammation and improve neural function. In addition, these effects may be related to inhibiting NLRP3 inflammasome-associated inflammation and reducing BCSB damage.

## 2. Materials and Methods

### 2.1. Animals and Spinal Cord Injury Model

We purchased 200–250 g male rats from SLAC Laboratory Animal Co., Ltd. (Shanghai, China). The rats were kept in a constant temperature and humidity environment and had free access to food and water. All procedures were in accordance with guidelines developed by the National Institutes of Health of China. To establish a model of t-SCI, the rat T10 spine was exposed by surgical laminectomy, and the spinal cord was clamped with a vascular clamp (30 g force, INS 14120, Kent Scientific, Torrington, CT, USA) for 30 s. Rats in the sham group received the same operation without the clamp.

### 2.2. Drug and Small Interfering RNA Administration

Sodium benzoate (SB) (100 mg/kg, Sigma-Aldrich, St. Louis, MO, USA) was diluted with 100 μL water and administered by gavage 1 h after t-SCI. DJ-1 mRNA mixtures, SCOS1 mRNA mixtures, and scramble siRNA were obtained from Thermo Fisher Scientific (Waltham, MA, USA). Entranster TM-in vivo transfection reagent delivered the siRNA according to the manufacturer’s recommendations. Intrathecal injection of siRNA solution was performed 48 h before operation. The needle tip was punctured into the subarachnoid space between L5 and L6 at a rate of 2 μL/min. The rats in the sham group underwent the same procedure but were not injected with the drug.

### 2.3. Experimental Procedures

#### 2.3.1. Experiment 1

To detect changes in expression and distribution of DJ-1 after t-SCI, the rats were randomly divided into the sham group and t-SCI groups with different time points (3 h, 6 h, 12 h, 24 h, 48 h, and 72 h) (*n* = 6). The location of DJ-1 was assessed by double immunofluorescence staining in the sham and t-SCI 24 h groups (*n* = 6).

#### 2.3.2. Experiment 2

To investigate the neuroprotective function of DJ-1 after t-SCI, the DJ-1 siRNA and selective DJ-I agonist SB were introduced. Rats were randomly distributed into the sham and t-SCI groups with vehicle, scramble siRNA, DJ-1 siRNA, SB, and DJ-1 siRNA combined with SB, respectively (*n* = 6). Western blotting was carried out to assess the expression of DJ-1, NLRP3, cleaved caspase 1, IL-1β, IL-18, MMP-9, Claudin-5, Occludin, and ZO-1 (*n* = 6).

#### 2.3.3. Experiment 3

To examine the effect of DJ-1 on neurological function, the Basso, Beattie, and Bresnahan (BBB) score and inclined plane test (IPT) were determined after t-SCI (*n* = 6). Rats were divided into the sham group and t-SCI groups with vehicle, SB, and DJ-1 siRNA, respectively (*n* = 6).

#### 2.3.4. Experiment 4

To further analyze the pathway of DJ-1-mediated NLRP3-related inflammation, we introduced the SOCS1 siRNA to inhibit the expression of SOCS1. Rats were assigned into five groups: sham group and t-SCI with vehicle, SB, SOCS1 siRNA, and SB + SOCS1 siRNA, respectively (*n* = 6). The expressions of DJ-1, SOCS1, and NLRP3 were detected by Western blotting in each group (*n* = 6). Rac1 activities and ROS levels were detected in these groups, respectively. The experimental design and animal groups are shown in Figure 1.

### 2.4. Long-Term Neurological Function Analysis

The BBB score was used to evaluate hind limb locomotor function. The score range is from 0 to 21; higher scores mean poorer motor function [22]. The IPT was investigated using a board secured at one end with the free edge of the board gradually raised to increase the angle of the incline. The maximum angle at which the rats maintained stability for at least 5 s was recorded as the inclined plane test angle. These two tests were conducted at 3, 7, 14, and 21 days following surgery.

### 2.5. Western Blot

We used an NE-PER Nuclear and Cytoplasmic Extraction Kit (Thermo, Rockford, IL, USA) to extract nuclear and cytoplasmic proteins.

Equal amounts of protein from samples were loaded and separated by sodium dodecyl sulfate (SDS)-polyacrylamide gel electrophoresis and transferred onto PVDF membranes (Bio-RAD Laboratories, Hercules, CA, USA). Firstly, the membrane and the corresponding primary antibody of the protein to be measured were incubated overnight at 4 °C. Then, the secondary antibody (1:10,000, Zhongshan Jinqiao ZB-2301 or ZB-2305) was incubated at room temperature for 1 h with an ECL kit (Thermo Scientific, Waltham, MA, USA). The band density of observed proteins was quantitatively analyzed by Image J software (NIH).

### 2.6. Cellular Immunofluorescence

A 0.5 cm segment of the spinal cord was taken from the center of the injury site. An axial frozen section of the spinal cord 2 mm from the center of the injury site (20 μm) was taken and incubated with anti-DJ1 (1:125, AB76008, Abcam, Waltham, MA, USA) and anti-IBa1 (1:500, Abcam AB5076) antibodies at 4 °C overnight, respectively. Following this, the samples were reacted with secondary antibodies (1:500, Invitrogen, Thermo Fisher Scientific, Waltham, MA, USA) at 25 °C for 2 h. Finally, DAPI (1μg/mL, Roche Inc., Basel, Switzerland) was used for dyeing the nucleus and mounting. A fluorescence microscope and Photoshop 13.0 software (Adobe Systems Inc., San Jose, CA, USA) were used to observe the tissue sections and for photograph post-processing. Six sections were obtained from each sample, and one random grey matter field per section was used to count the cell numbers at 200× magnification. The DJ-1 expression was expressed as the mean ratio of DJ-1-positive cells to total cells in each group. Neuroinflammation was evaluated by the proportion of Iba1-positive cells in each group.

### 2.7. Rac1 Activation Assay

Rac1 activity was detected using a Rac1 Activation Assay Kit (ab211161, Abcam) according to the product instruction. GST-PBD (p21-binding domain of PAK) was used for the Rac1 activity assays. The CRC cells were transfected with DMTN overexpression, knockdown, or control vector and washed with PBS after 72 h. Then, the cells were lysed in ice-cold Mg^2+^ lysis buffer and centrifuged for 5 min at 13,000× *g* at 4 °C. Next, 40 μL of supernatant was taken to determine the total Rac1 levels. The remaining supernatants were incubated with GST-PBD on glutathione-sepharose beads and rotated at 4 °C for 2 h. The beads were washed extensively in lysis buffer, and the bound proteins were separated by SDS-PAGE and then immunoblotted with anti-Rac1 antibodies.

### 2.8. Measurement of ROS Levels

A ROS assay kit from Nanjing, China was used to detect the ROS level according to the protocol of the manufacturer. A 0.5 cm-long spinal cord sample was taken from the center of the injured site and infiltrated with 0.1 mol/L PBS, and then the samples were weighed and homogenized in PBS (1 g: 20 mL). After that, the mixtures were centrifuged at 1000× *g* for 10 min at 4 °C. The supernatant (190 μL) and DCFH-DA (10 μL, 1 mol/L) were added to 96-well plates and, as a control, the same volume of PBS was mixed with the supernatant. Then, the supernatant was detected at an emission wavelength of 525 nm and excitation wavelength of 500 nm via a strip reader (Biotek Instruments Inc., Winooski, VT, USA). The ROS levels were displayed in the form of fluorescence/mg protein.

### 2.9. Statistical Analysis

Data were represented as mean ± standard deviation (SD). Statistical analysis was carried out with SPSS (version 25.0) and GraphPad Prism for Windows (GraphPad, Inc., San Diego, CA, USA). One-way analysis of variance (ANOVA) was adopted to analyze the statistical differences among groups. *p* < 0.05 was considered to be statistically significant.

## 3. Results

### 3.1. Temporal Patterns and Localization of DJ-1 after t-SCI

Western blotting indicated that the level of DJ-1 began to increase significantly at 3 h and peaked at 24 h post-injury compared with the sham group. The protein levels of DJ-1 significantly decreased after 24 h post-injury but were still higher than those in the sham group (Figure 2A). Furthermore, DJ-1 was mainly located in neurons (Figure 2B). The proportion of DJ-1 positive neurons was increased at 24 h post-injury compared with the sham group (Figure 2C,D).

### 3.2. Downregulation of DJ-1 Increased NLRP3 Inflammasome Activation and Destruction of BCSB

Given that cleaved caspase 1, IL-1β, and IL-18 are common downstream molecules of the NLRP3 inflammasome and neuroinflammation indicators, these levels were examined after upregulating and downregulating DJ-1. Western blotting showed that the protein levels of DJ-1, NLRP3, cleaved caspase 1, IL-1β, and IL-18 were significantly increased after t-SCI compared to the sham group. Treatment with DJ-1 siRNA significantly decreased DJ-1 and elevated the NLRP3. The level of NLRP3-related downstream molecules (cleaved caspase 1, IL-1β, IL-18) exhibited an increase, indicating that the downregulation of DJ-1 could activate NLRP3 inflammasome and aggravate neuroinflammation (Figure 3A–F).

Previous studies have demonstrated that neuroinflammation might lead to the destruction of BCSB, which is usually accompanied by the degradation of tight junction proteins including Claudin-5, Occludin, and ZO-1. Our results showed that t-SCI promoted MMP-9 expression and reduced tight junction proteins, which means the destruction of BCSB, whereas, after downregulating DJ-1, the expression of MMP-9 was higher, and the tight junction proteins were lower than the t-SCI + scramble siRNA (Figure 3G–J).

### 3.3. SB Increased the Neuroprotective Effects of DJ-1 on Inhibiting NLRP3 Inflammasome Activation and Alleviating BCSB Disruption

Conversely, treatment with SB significantly increased DJ-1 and decreased the NLRP3, cleaved caspase 1, IL-1β and IL-18, compared to the t-SCI + vehicle group. Furthermore, tight junction proteins were higher, and MMP-9 was lower in the t-SCI + SB group. The results indicated that SB increased the neuroprotective effects of DJ-1 on mediating NLRP3-related neuroinflammation and BCSB disruption. Additionally, DJ-1 siRNA can reverse the effect of SB when the protein levels display no obvious differences between the t-SCI + vehicle and t-SCI + SB + DJ-1 siRNA groups (*p* > 0.05, Figure 4A–G).

### 3.4. DJ-1 Improved Long-Term Neurological Function

In this part of this study, blinded BBB scores and the IPT results were used to evaluate the locomotion function. Throughout post-SCI recovery, BBB scores were higher in the t-SCI + SB group than t-SCI + vehicle at each time point and were significantly higher at the 21st and 28th day post-injury (*p* < 0.05, Figure 5A). The angles of incline were bigger in the t-SCI + SB group than t-SCI + vehicle and showed significant change on the 28th day post-injury (*p* < 0.05, Figure 5B). The results showed that the SB treatment group experienced much greater functional recovery after t-SCI. This demonstrates that upregulating DJ-1 could promote long-term neurological function rather than short-term, whereas DJ-1 reduction on neuromotor function was not significant.

### 3.5. DJ-1 Mediated Neuroinflammation through SOCS1

The next part of this study was conducted to determine whether the effects of DJ-1 mediating NLRP3 inflammasome in the pathological process following t-SCI were exerted through SOCS1. SOCS1 siRNA was introduced to inhibit SOCS1. Western blotting indicated that DJ-1 and SOCS1 were higher after SB treatment than in the t-SCI + vehicle group, whereas the administration of SOCS1 siRNA increased the DJ-1, possibly through a negative feedback mechanism, indicating that SOCS1 could act as a downstream molecule of DJ-1 (Figure 6A–C).

The results revealed that SB decreased Rac1-GTP, ROS, and NLRP3; however, these effects were reversed by SOCS1 siRNA treatment (t-SCI + SB + SOCS1 siRNA vs. t-SCI +SB, Figure 6D–F). We also observed that reducing SOCS1 could eliminate the neuroprotective effect of DJ-1 since the level of Rac1-GTP, ROS, and NLRP3 inflammasome did not obviously alter in the t-SCI + SB + SOCS1 siRNA group (Figure 6D–F). The evidence above suggests that DJ-1 plays a role in mediating NLRP3 inflammasome-related neuroinflammation via elevating the protein levels of SOCS1.

## 4. Discussion

In this study, we made the following major finds: (1) the expression of DJ-1 was increased after t-SCI and was mainly located in neurons. (2) The knockdown of DJ-1 with specific siRNA significantly activated NLRP3 and its associated inflammatory cytokines, increased MMP-9, and reduced tight junction proteins, which resulted in increased neuroinflammation and destruction of BSCB integrity. (3) The up-regulation of DJ-1 reversed these effects and also improved neurological locomotor function. (4) The use of SOCS1 siRNA abolished the neuroprotective effects of DJ-1 induced by SB and increased the levels of ROS, Ras1, and NLRP3. The working model for this study is available in Figure 7.

Regulating post-injury inflammation to improve neurological prognosis is a hot topic in the field of spinal cord injury [23]. Methylprednisolone sodium succinate was one of the first drugs used clinically to prevent secondary damage, by stabilizing cell membranes and providing anti-inflammatory effects, but with clinical efficacy and unavoidable side effects [24]. There have been many subsequent clinical studies targeting multiple drugs including COX inhibitors [25], corticosteroids [26], and minocycline [27], but none of them have achieved satisfactory results. The underlying reason is that the cognition of the mechanism of inflammation is not comprehensive and clear, and appropriate targets and drugs have not been found.

DJ-1 is a small, ubiquitously expressed protein in the brain that exists as a homodimer in the cytoplasm, mitochondria, and nucleus [28]. The DJ-1 gene has been identified as the pathogenic gene in familial Parkinson’s disease and was later found to show neuroprotective effects in neurodegenerative diseases [29,30]. The suggested neuroprotective mechanisms of DJ-1 include reducing neuronal oxidative stress and attenuating microglial activation [31,32,33]. Our previous experiments found that DJ-1 could decrease oxidative stress related to nerve injury after t-SCI [34]. However, there are few studies on the mechanism of DJ-1 mediating neuroinflammation in spinal cord injury. Our results indicate that DJ-1 increased after injury and peaked at 24 h and was involved in the acute injury stage. Furthermore, the upregulation of DJ-1 decreased inflammatory cytokines, including IL-18, IL-1β, and ca caspase-1, and improved long-term neuromotor function.

To further analyze the potential mechanisms of DJ-1 mediating the inflammation, we explored the activation of the NLRP3 inflammasome, which controls the maturation and release of pro-inflammatory cytokines, especially IL-18, IL-1β, and caspase-1 [35]. The NLRP3 inflammasome is the most characteristic in the NLR family and was found to participate in neurodegenerative diseases and cerebrovascular diseases [36,37,38]. A previous study found that NLRP3 inflammasome significantly increased after SCI [39], and that suppressing NLRP3 inflammasome activation could alleviate SCI-induced neuroinflammation and spinal nerve injury [40]. We found that the protein expressions of NLRP3, caspase-1, IL-1β, and IL-18 were increased after SCI, indicating that the NLRP3 inflammasome was activated. Downregulating DJ-1 reduced the activation of the NLRP3 inflammasome, and SB reversed this effect. A previous study showed that the suppressor of cytokine signal 1 (SOCS1) inhibited the activity of NLRP3 [41]. SOCS1 is a key physiological suppressor of cytokine found to weaken cytokine signals, usually via the NF-κB, JAK2, and TLR4 pathways [42], as well as regulating neuronal immunity in CNS cells [43]. SOCS1 also participates in oxidative stress to degrade active Rac1 and inhibit the production of reactive oxygen species (ROS) [44], which is a vital trigger for the activation of the NLRP3 inflammasome [45]. Consistent with our predictions, the inhibition of SOCS1 increased DJ-1 in feedback, which acted as an upstream of SOCS1, and inhibited the activation of the NLRP3 inflammasome by inhibiting ROS.

Additionally, BSCB destruction is the main pathological change after t-SCI, and results in a poor prognosis [46]. The regulatory and protective functions of the BSCB stem from a highly evolved, complex network of tight junction proteins, including ZO-1, Occludin, and Claudin-5 [47,48]. After t-SCI, tight junction proteins degrade and BSCB permeability increases [49], resulting in leukocyte infiltration and inflammatory cascade reaction [50]. MMP-9 is a gelatinase secreted by infiltrating neutrophils and is also involved in this process, acting as a key mediator of early inflammation [51]. After t-SCI, neutrophils infiltrate and release the MMP-9, which degrades the extracellular matrix, tight junction proteins and surrounding substrate composition [52]. In our study, MMP-9 was in a higher condition, and tight junction proteins (Claudin-5, Occludin and ZO-1) were in a lower condition after t-SCI. Furthermore, the upregulation of DJ-1 alleviated BCSB damage by decreasing MMP-9 and increasing tight junction proteins. As also seen in previous research, activating signaling pathways and releasing inflammatory cytokines after the activation of the NLRP3 inflammasomes creates a pro-inflammatory environment [53], which will mediate MMP-9 and tight-junction protein expression and promote BSCB destruction [54].

Through our study, the potential role of DJ was preliminarily explored, which provided some theoretical reference for further clinical transformation. However, there are several weaknesses in our study. DJ-1 has been proven to exhibit neuroprotective effects after t-SCI through multiple mechanisms, but we only focused on NLRP3 inflammasome-related neuroinflammatory. The underlying mechanisms of how DJ-1 reduces BSCB distribution require more investigation, and in vivo studies should be completed in future research.

## 5. Conclusions

In conclusion, we have shown a previously uncharacterized signaling pathway in which DJ-1 played a neuroprotective role after t-SCI. Our data indicated that DJ-1 suppressed t-SCI induced neuroinflammation by inhibiting the activation of NLRP3 inflammasome via the SOCS1/Rac1/ROS pathway. Furthermore, the pharmacological upregulation of DJ-1 alleviated inflammasome-related BSCB destruction and promoted long-term neurological locomotor function. Altogether, DJ-1 might be a promising therapeutic target for t-SCI, and further studies are needed.

## Figures and Tables

**Figure 1 jcm-11-03716-f001:**
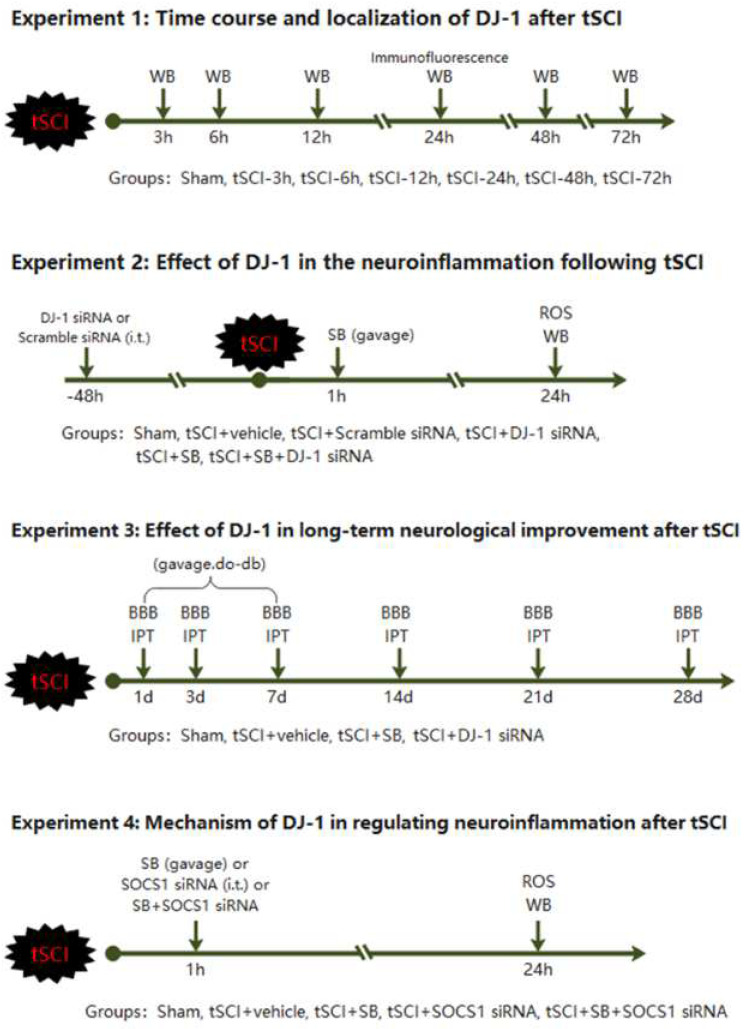
Experimental design and animal groups.

**Figure 2 jcm-11-03716-f002:**
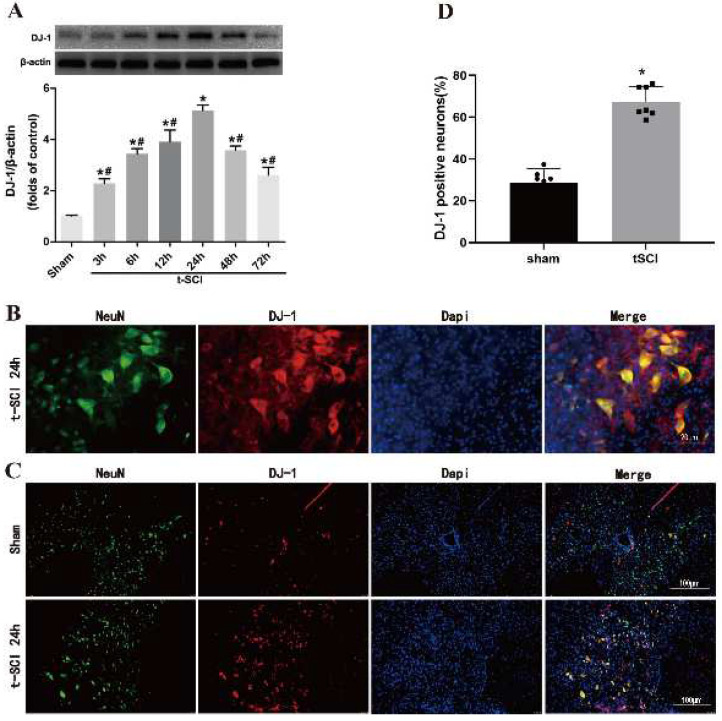
Temporal patterns and localization of DJ-1 after t-SCI. (**A**) Representative Western blotting of DJ-1 after t-SCI. (**B**) Representative immunofluorescence double staining micrographs showing the localization of S DJ-1. (**C**,**D**) The proportion of DJ-1-positive neurons in sham groups and after t-SCI. *n* = 6 for each group. * *p* < 0.05 vs. *sham*; # *p* < 0.05 vs. *t-SCI 24 h*.

**Figure 3 jcm-11-03716-f003:**
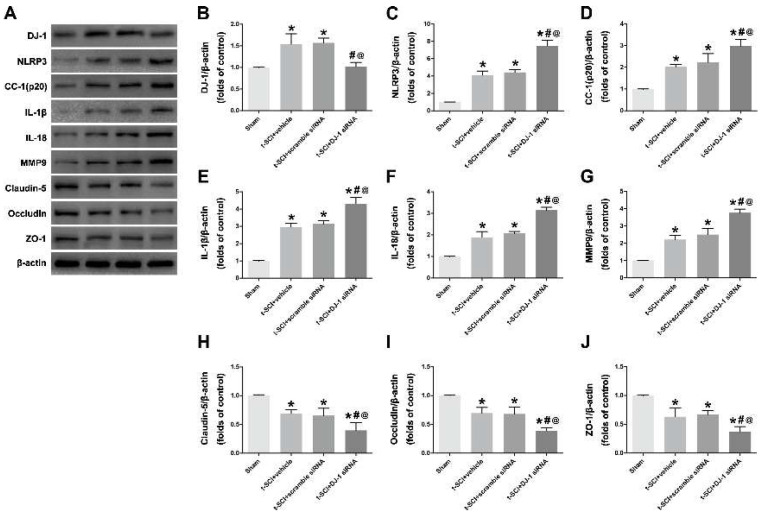
Downregulation of DJ-1 increased NLRP3 inflammasome activation and destruction of BCSB. (**A**–**G**): Representative Western blotting images and quantitative analyses of DJ-1, NLRP3, cleaved caspase 1 (CC-1), IL-1β, IL-18 and MMP9 expression at 24 h post-injury. (**H**–**J**): Representative Western blotting images and quantitative analyses of tight junction proteins including Claudin-5, Occludin and ZO-1 at 24 h post-injury. *n* = 6 for each group. * *p* < 0.05 vs. *sham*; # *p* < 0.05 vs. *t-SCI + vehicle*; @ *p* < 0.05 vs. *t-SCI + scramble siRNA*.

**Figure 4 jcm-11-03716-f004:**
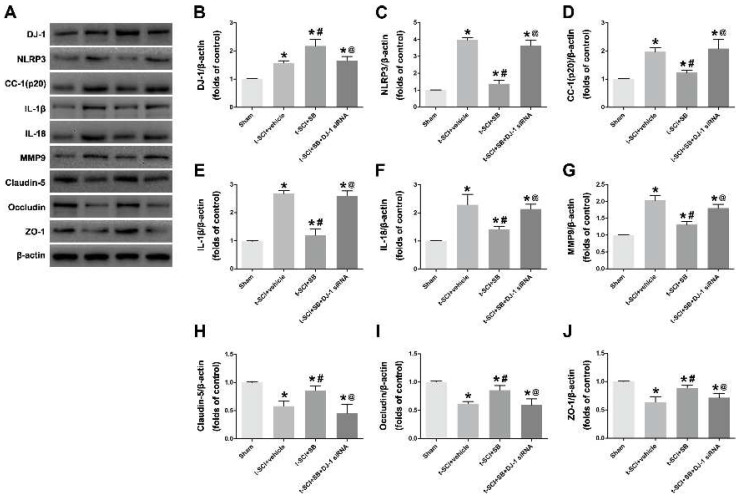
SB increased the neuroprotective effects of DJ-1 on inhibiting NLRP3 inflammasome activation and alleviating BCSB disruption. (**A**–**G**): Representative Western blotting images and quantitative analyses of DJ-1, NLRP3, cleaved caspase 1 (cc-1), IL-1β, IL-18 and MMP9 expression at 24 h post-injury. (**H**–**J**): Representative Western blotting images and quantitative analyses of tight junction proteins including Claudin-5, Occludin and ZO-1 at 24 h post-injury. *n* = 6 for each group. * *p* < 0.05 vs. *sham*; # *p* < 0.05 vs. *t-SCI + vehicle*; @ *p* < 0.05 vs. *t-SCI + SB*.

**Figure 5 jcm-11-03716-f005:**
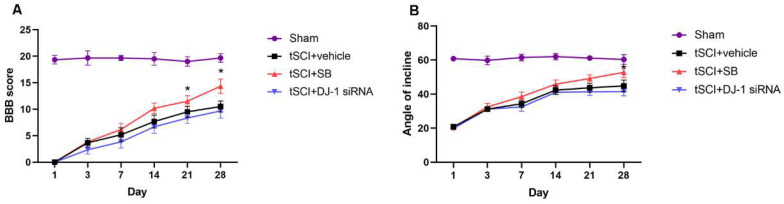
Effect of DJ-1 on neurological function. (**A**). The Basso, Beattie, and Bresnahan (BBB) scores were higher in the t-SCI + SB group at 21 and 28 days. (**B**). The angle of the incline in the inclined plane test (IPT) was bigger in the t-SCI + SB group at days 28 post-injury. *n = 6 for each group*. * *p* < 0.05 vs. *t-SCI + vehicle*.

**Figure 6 jcm-11-03716-f006:**
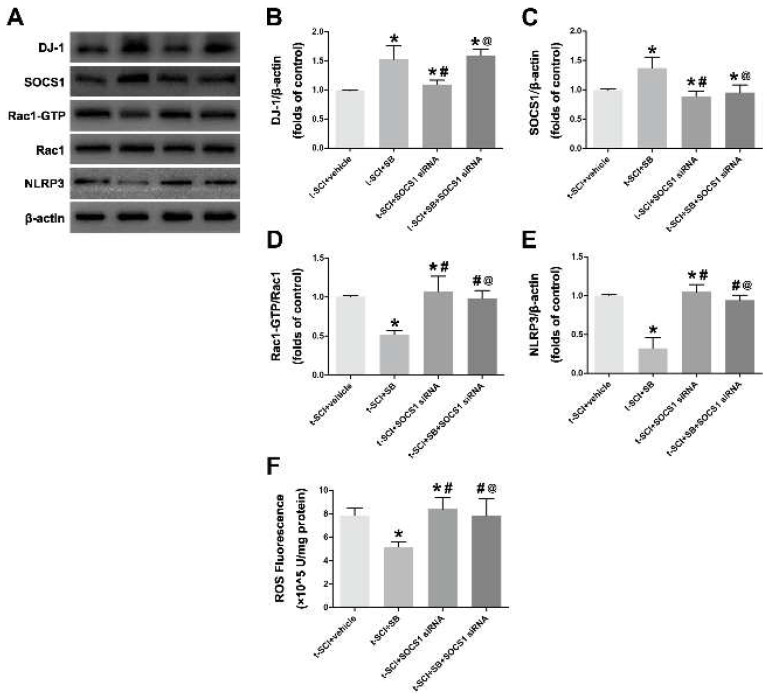
DJ-1 mediated NLRP3-related neuroinflammation through SOCS1. (**A**–**E**): Representative Western blotting images of DJ-1, SOCS1, Rac-GTP and NLRP3. (**F**). Quantification of ROS. * *p* < 0.05 vs. *t-SCI + vehicle*; # *p* < 0.05 vs. *t-SCI + SB;* @ *p* < 0.05 vs. *t-SCI + SOCS1 siRNA*.

**Figure 7 jcm-11-03716-f007:**
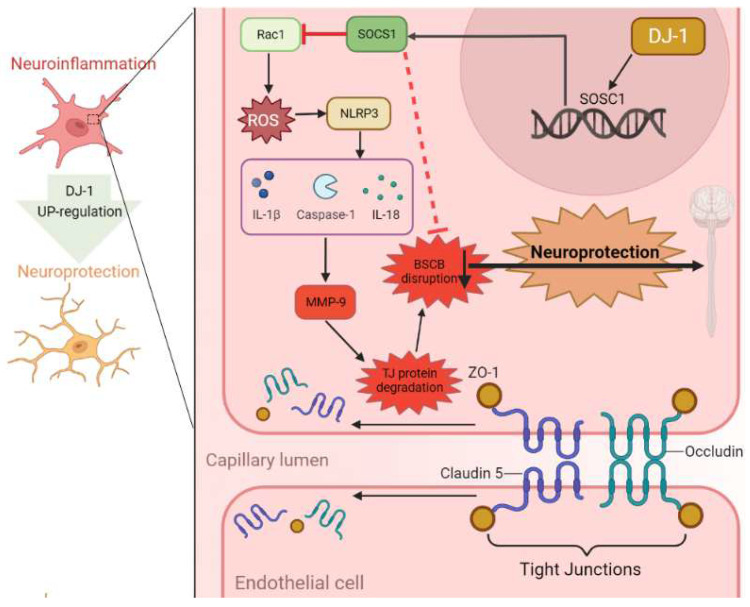
Schematic mechanism of neuroprotection of DJ-1 in traumatic spinal cord injury. DJ-1 alleviates neuroinflammation and the related blood-spinal cord barrier destruction by suppressing NLRP3 inflammasome activation via SOCS1/Rac1/ROS pathway.

## Data Availability

The data that support the findings of this study are available from the corresponding author upon reasonable request.

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
