# Peer review of "DJ-1 Alleviates Neuroinflammation and the Related Blood-Spinal Cord Barrier Destruction by Suppressing NLRP3 Inflammasome Activation via SOCS1/Rac1/ROS Pathway in a Rat Model of Traumatic Spinal Cord Injury"

_jcm, 2022, doi:10.3390/jcm11133716_

Round 1

Reviewer 1 Report

The authors propose their animal model study to show uncharacterized signaling pathway in which DJ-1 plays a neuroprotective role after traumatic SCI. They found DJ-1 to be a promising therapeutic target for tSCI

The manuscript is overall well written, figures are nice, and I only have a few minor comments:

Please mention the animal study/rat model in the title.

Traumatic SCI is not fatal in the majority of cases. Please correct in the introduction.

Line 40: Introduce abbreviations only once.

Reviewer 2 Report

THE authors performed a study of DJ-1 gene expression commonly expressed in Parkinson's disease, including in spinal cord neurotrauma. The study turns out to be interesting from the topic point of view however it is in large parts difficult to understand.

The importance of DJ-1 is not mentioned in the abstract.

What are the rationales for Vascular clamps in the rat being an expression of inflammatory damage to the cord? Wouldn't it be useful to compare with direct spinal cord injury?

The results appear to be speculative and analyze little of the data obtained

Clinical references are lacking in the discussion.

The discussion appears to be difficult to follow.

A major revision of the English is needed.

Round 2

Reviewer 2 Report

Now is more readable and acceptable